# Generating High Fidelity Synthetic Data
# via Coreset selection and Entropic Regularization

## Abstract

Generative models have the ability to synthesize data points drawn from the data distribution, however, not all generated samples are high quality. In this paper, we propose using a combination of coresets selection methods and "entropic regularization" to select the highest fidelity samples. We leverage an Energy-Based Model which resembles a variational auto-encoder with an inference and generator model for which the latent prior is complexified by an energy-based model. In a semi-supervised learning scenario, we show that augmenting the labeled data-set, by adding our selected subset of samples, leads to better accuracy improvement rather than using all the synthetic samples.

## 1   Introduction

In machine learning, augmenting data-sets with synthetic data has become a common practice which potentially provides significant improvements in downstream tasks such as classification. For example, in the case of images, recent methods like MixMatch, FixMatch and Mean Teacher [1] [12] [13] have proposed data augmentation techniques which rely on simple pre-defined transformations such as cropping, resizing, etc.

However, generating augmentations is not as straightforward in all modalities. Hence, one suggestion is to use samples from generative models to augment the data-sets. One issue that arises is that simply augmenting a data-set using a generative model can often lead to the degradation of classification accuracy due to some poor samples drawn from the generator. The question arises: can we filter the lower quality generated samples to avoid degradation in accuracy? In our method we select a subset of synthetic samples which have high fidelity to the underlying data-set via CRAIG [6], additionally we introduce "entropic regularization" by filtering samples with low entropy over the latent classifier.

In semi-supervised learning, the goal is to learn a classifier model which maintains high classification accuracy while reducing the number of labeled observed examples. Generative modeling and especially likelihood-based learning is a principled formulation for unsupervised and semi-supervised learning. Within this family of models, energy-based models (EBM) are particularly convenient for semi-supervised learning, as they may be interpreted as generative classifiers. That is, we not only have access to the class predictions but may also draw samples from the model.

Another direction in supervised learning is to reduce the amount of computation involved in training a model by reducing the data-set to a smaller subset. Such sets are coined *coresets* as a smaller set of representative points attempts to approximate the geometry of a larger point set under some metric. Recent art [6] introduces a novel algorithm CRAIG which constructs a weighted coreset such that the gradient over the full training data-set is closely estimated, which allows for gradient descent on the smaller coreset with considerable improvement in the sample- and computational-efficiency.

Submitted to NeurIPS 2022 Workshop on Synthetic Data for Empowering ML Research. Do not distribute.

In this work, we show that semi-supervised learning and coreset subset selection are complementary and improve generalization as well as generation quality. First, a generative classifier is learned on a large set of unlabelled data and a small set of labeled data pairs. Then, the generative model is utilized to draw class conditional samples which augment the labeled data pairs. As such augmentation might be a considerably large set, in fact, we can draw infinite samples from the generative model, we recruit CRAIG to reduce the conditional samples to a much smaller coreset while approximately maintaining the full gradient over the cross-entropy term. As the generative model might synthesize conditional samples of low quality or even incorrect class identity, we apply an entropic filter to remove noisy samples. By learning a joint generative classifier we learn a generator that can produce samples that improve classification accuracy as well as a classifier that can boost generative capacity and quality.

This method may be interpreted as a learned (and filtered) data augmentation as opposed to classical data augmentation in which the set of augmentation functions (e.g., convolution with Gaussian noise, horizontal or vertical flipping, etc.) is pre-defined and could be specific to a data-set or modality. We demonstrate the efficacy of the method by a significant improvement in classification performance.

## 2  Synthetic Data Generation for Semi-Supervised Learning

**Notation**   Let $x \in \mathcal{R}^D$ be an observed example. Let $y$ be a $K$-dimensional one-hot vector as the label for classification with $K$ categories. Suppose $\mathcal{L} = \{(x_i, y_i) \in \mathbb{R}^D \times \{k\}_{k=1}^K, i = 1, ..., M\}$ denotes a set of labeled examples where $K$ indicates the number of categories and $\mathcal{U} = \{x_i \in \mathbb{R}^D, i = M + 1, ..., M + N\}$ denotes a set of unlabeled examples.

**Semi-Supervised Learning**   Let $p_\theta(y \mid x)$ denote a soft-max classifier with parameters $\theta$. The goal of semi-supervised learning is to learn $\theta$ with "good" generalization while decreasing the number of labeled examples $M$.

### 2.1  Latent Energy Based Model

Let $z \in \mathbb{R}^d$ be the latent variables, where $D \gg d$. We assume a Markov chain $y \to z \to x$. Then the joint distribution of $(y, z, x)$ is

$$p_\theta(y, z, x) = p_\alpha(y, z)\, p_\beta(x|z), \tag{1}$$

where $p_\alpha(y, z)$ is the prior model with parameters $\alpha$, $p_\beta(x|z)$ is the top-down generation model with parameters $\beta$, and $\theta = (\alpha, \beta)$. Then, the prior model $p_\alpha(y, z)$ is formulated as an energy-based model [10],

$$p_\alpha(y, z) = Z(\alpha)^{-1} \exp(F_\alpha(z)[y])\, p_0(z). \tag{2}$$

where $p_0(z)$ is a reference distribution, assumed to be isotropic Gaussian. $F_\alpha(z) \in \mathbb{R}^K$ is parameterized by a multi-layer perceptron. $F_\alpha(z)[y]$ is $y_{\text{th}}$ element of $F_\alpha(z)$, indicating the conditional negative energy. $Z(\alpha)$ is the partition function. In the case where the label $y$ is unknown, the prior model $p_\alpha(z) = \sum_y p_\alpha(y, z) = Z(\alpha)^{-1} \sum_y \exp(F_\alpha(z)[y]) p_0(z)$. Taking log of both sides:

$$\log p_\alpha(z) = \log \sum_y \exp(F_\alpha(z)[y]) + \log p_0(z) - \log Z(\alpha), \tag{3}$$

The prior model can be interpreted as an energy-based correction or exponential tilting of the reference distribution, $p_0$. The correction term is $F_\alpha(z)[y]$ conditional on $y$, while it is $\log \sum_y \exp(F_\alpha(z)[y])$ when $y$ is unknown. Denote

$$f_\alpha(z) = \log \sum_y \exp(F_\alpha(z)[y]), \tag{4}$$

and then $-f_\alpha(z)$ is the free energy [2]. The soft-max classifier is $p_\alpha(y|z) \propto \exp(\langle y, F_\alpha(z) \rangle) = \exp(F_\alpha(z)[y])$.

The generation model is the same as the top-down network in VAE [4], $x = g_\beta(z) + \epsilon$, where $\epsilon \sim \mathrm{N}(0, \sigma^2 I_D)$, so that $p_\beta(x|z) \sim \mathrm{N}(g_\beta(z), \sigma^2 I_D)$.

73  We use variational inference to learn our latent space EBM by minimizing the evidence lower bound
74  (ELBO) over our energy, encoder, and generator models jointly. Refer to appendix B for more details
75  about learning the model.

76  In summary, we can use the above model to i) classify data points ii) generate class-conditional
77  samples iii) compute entropy for each generated sample. We will leverage these properties in the
78  later sections to get better augmentation for our data-set.

## 2.2  Sampling Synthetic data from the EBM

80  Naturally, increasing the cardinality of the set of labeled samples $\mathcal{L}$ may improve the classification
81  accuracy of soft-max classifier $p_\theta(y \mid x)$. In the case of image models, traditional methods recruit a set
82  of transformations or permutations of $x$ such as convolution with Gaussian noise or random flipping.
83  Instead we leverage the learned top-down generator $p_\beta(x|z)$ to augment $\mathcal{L}$ with class conditional
84  samples. This is beneficial as (1) the generative path is readily available as an auxiliary model of
85  learning the variational posterior $q_\phi(z|x)$ by auto-encoding variational Bayes, (2) hand-crafting of
86  data augmentation is domain and modality-specific, and (3) in principle the number of conditional
87  ancestral samples is infinite and might capture the underlying data distribution well.

88  We may construct the augmented set of $L$ labelled samples $\mathcal{L}^+ = \{(x_i, y_i)\}$ by drawing conditional
89  latent samples from the energy-based prior model $p_\alpha(y, z)$ in the form of Markov chains. Then, we
90  obtain data space samples by sampling from the generator $p_\beta(x|z)$.

91  First, for each label $y$, we draw an equal number of samples $\mathcal{Z} = \{z_i\}$ in latent space. One convenient
92  MCMC is the overdamped Langevin dynamics, which we run for $T_{LD}$ steps with target distribution
93  $p_\alpha(y, z)$,

$$z \sim p_0(z), \tag{5}$$

$$z_{t+1} = z_t + s\nabla_z \left[ f_\alpha(z)[y] - \|z\|^2 / 2 \right] + \sqrt{2s}\epsilon_t, \ t = 1, \ldots, T_{LD} \tag{6}$$

94  with negative conditional energy $f_\alpha(z)[y]$, discretization step size $s$, and isotropic $\epsilon_t \sim N(0, I)$.

95  Then, we draw conditional samples $\{x_i\}$ in data space given $\{z_i\}$ from the top-down generator
96  model $p_\beta(x|z)$,

$$\mathcal{L}^+ = \{(x_i \sim p_\beta(x|z_i), y_i) \mid i = M + N, \ldots, M + N + L\} \tag{7}$$

97  which results in an augmented data-set of $L$ class conditional samples.

## 2.3  Entropic Regularization

99  When learning the generative classifier on both labelled samples $\mathcal{L}$ and the above naive construction
100 of augmentation $\mathcal{L}^+$, the classification accuracy tends to be worse than solely learning from $\mathcal{L}$.
101 As depicted in Figure 1a, a few conditional samples suffer from either low visual fidelity or even
102 incorrect label identity. This reveals the implicit assumption of our method is that $p_\beta(x|z)p_\alpha(z|y)$ is
103 reasonable "close" to the true class conditional distribution $p(x|y)$ under some measure of divergence,
104 which is not guaranteed.

105 To address the issue of outliers, we propose to exclude conditional samples for which the entropy in
106 logits $\mathcal{H}(p_\theta(y|z))$ exceeds some threshold $\mathcal{T}$. We propose the following criteria for outlier detection,

$$\mathcal{H}(z) = -\sum_y p_\theta(z|y) \log p_\theta(z|y). \tag{8}$$

107 Note,(8) is the classical Shannon entropy of over the soft-max normalized logits of the classifier.
108 Then, we may construct a more faithful data augmentation as follows,

$$\mathcal{Z}_\mathcal{T} = \{z_i \sim p(z|y_i) \mid \mathcal{H}(z_i) < \mathcal{T}, i = M + N, \ldots, M + N + L\}, \tag{9}$$

$$\mathcal{L}_\mathcal{H}^+ = \{(x_i \sim p_\beta(x|z_i), y_i) \mid i = M + N, \ldots, M + N + L\}. \tag{10}$$

109 Figure 1b depicts conditional samples sorted by $\mathcal{H}(z)$ for which samples with relatively large Shannon
110 entropy suffer from low visual fidelity.

111 The learning and sampling algorithm is described in Algorithm 1 (appendix) as an extension of [10].

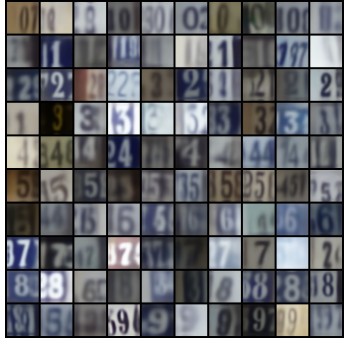
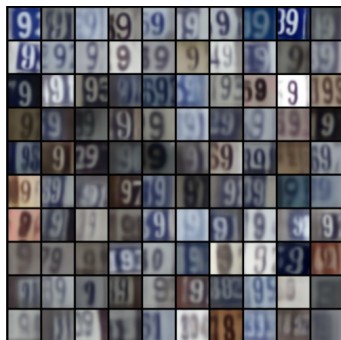

(a) Unsorted Conditional Samples.     (b) Sorted Conditional Samples.

Figure 1: Class conditional samples drawn from $p_\beta(x|z)p_\alpha(z|y)$. (a) Outliers suffer from low visual fidelity (e.g., the last sample in the row of "ones") or wrong label identiy (e.g., the last image of row of "sevenths". (b) Conditional samples sorted by increasing Shannon entropy $\mathcal{H}(z)$ over the logits.

## 2.4 Coreset Selection

Training machine learning models on large data-sets incur considerable computational costs. There has been substantial effort to develop subset selection methods that can carefully select a subset of the training samples that generalize on par with the entire training data [6] [11]. Since we can generate virtually infinite amount of synthetic samples, we must select the best subset of points to augment our base data-set with. Intuitively CRAIG selects a subset that can best cover the gradient space of the full data-set. It does this by selecting exemplar medoids from clusters of datapoints in the gradient space. As a bi-product, CRAIG robustly rejects noisy and even poisoned datapoints. The subset corset algorithm ADACORE improves on CRAIG's results by selecting diverse subsets [11]. Utilizing coreset methods allows us to select samples from the generator that is representative of the ground truth data-set while rejecting points that may negatively impact our network performance.

Formally, the CRAIG [6] algorithm aims to identify the smallest subset $S \subset V$ and corresponding per-element stepsizes $\gamma_j > 0$ that approximate the full gradient with an error at most $\epsilon > 0$ for all the possible values of the optimization parameters $w \in \mathcal{W}$.

$$S^* = \arg\min_{S \subseteq V, \gamma_j \geq 0 \forall j} |S|, \text{ s.t. } \max_{w \in \mathcal{W}} \left\| \sum_{i \in V} \nabla f_i(w) - \sum_{j \in S} \gamma_j \nabla f_j(w) \right\| \leq \epsilon \tag{11}$$

For deep neural networks it is more costly to calculate the above metric than to calculate vanilla SGD, In deep neural networks, the variation of the gradient norms is mostly captured by the gradient of the loss w.r.t the inputs of the last layer $L$. [6] shows that the normed gradient difference between data points can be efficiently bounded approximately by

$$\|\nabla f_i(w) - \nabla f_j(w)\| \leq c_1 \left\| \Sigma'_L \left( z_i^{(L)} \right) \nabla f_i^{(L)}(w) - \Sigma'_L \left( z_j^{(L)} \right) \nabla f_j^{(L)}(w) \right\| + c_2 \tag{12}$$

where $z_i^{(l)} = w^{(l)} x_i^{(l-1)}$. This upper bound is only slightly more expensive than calculating the loss. In the case of cross entropy loss with soft-max as the last layer, the gradient of the loss w.r.t. the $i$-th input of the soft-max is simply $p_i - y_i$, where $p_i$ are logits and $y$ is the one-hot encoded label. As such, for this case CRAIG does not need a backward pass or extra storage. This makes CRAIG practical and scalable tool to select higher quality generated synthetic data points.

## 2.5 Implicit learned data augmentation

In the following, we will re-interpret the above explicit data augmentation and entropic regularization into an implicit augmentation which can be merged into a simple term of the learning objective function.

The assumed Markov chain underlying the model is $y \rightarrow z \rightarrow x$. Let $\hat{z} \sim q_\phi(z|x)$ denote the conditional sample $\hat{z}$ from the approximate posterior given an observation $x$. Let $\hat{y} \sim p_\theta(y|\hat{z})$ denote the predicted label for which the logits of $C$ classes are given as $F_\alpha(z) = (F_\alpha(z)[1], F_\alpha(z)[2], \ldots, F_\alpha(z)[C])$.

The factorization which recruits the log-sum-exp lifting (3) as exponential tilting of the the reference distribution $p_0(z)$ so that the conditional $p_\alpha(y|z)$ is defined, and, amortized inference (19) with variational approximation of the posterior $q_\phi(z|x)$. These conditional distributions allow us to express learned data augmentation as the chain,

$$y \overset{q_T(z|y)}{\to} z \overset{p_\theta(x|z)}{\to} x \overset{q_\phi(z|x)}{\to} \hat{z} \overset{F_\alpha(z)[y]}{\to} \hat{y}. \tag{13}$$

in which the conditional $z|y$ is given as a MCMC dynamics. Specifically, we define $q_T(z|y)$ as $K$-steps of an overdamped Langevin dynamics on the learned energy-based prior $\exp(F_\alpha(z)[y])p_0(z)$, which iterates

$$z_{k+1} = z_k + s\nabla_z \log p(z_k|y) + \sqrt{2Ts}\epsilon_k, \quad k = 0, \ldots, K-1, \tag{14}$$

with discretization step-size $s$, temperature $T$ and isotropic noise $\epsilon_k \sim N(0, I)$.

For the (labeled) data distribution $p_{\text{data}}$ the labels $y$ are known. For the data augmentation, we assume a discrete uniform distribution over labels $y \sim U\{1, C\}$. Then, we define augmentation of synthesized examples as the marginal distribution

$$p_{aug}(x) = E_y E_{z|y}[p(x|z)p(z|y)]. \tag{15}$$

Then, we may introduce an augmented data-distribution as the mixture of the underlying labeled data-distribution $p_{\text{data}}$ and the augmentation $p_{\text{aug}}$ and mixture coefficient $\lambda$,

$$p_\lambda(x) = \lambda p_{\text{data}}(x) + (1-\lambda)p_{\text{aug}}(x). \tag{16}$$

As we have access to $p_\theta(y|x) = E_{p_\theta(z|x)}p_\theta(y|z)$ and can extend the objective to minimize the KL divergence under the augmented data distribution such that the labels $y$ of (labeled) $p_{\text{data}}$ and $p_{aug}$ are recovered under the model,

$$E_{p_\lambda(x)}[KL(p(y|x)\|p(\hat{y}|x))]. \tag{17}$$

In information theory, the Kraft-McMillian theorem relates the relative entropy $KL(p\|q) = E_p[\log p/q]$ to the Shannon entropy $H(p)$ and cross entropy $H(p, q)$,

$$KL(p\|q) = H(p, q) - H(p). \tag{18}$$

In our case, the first term reduces to soft-max cross entropy over the (labeled) data distribution $p_{\text{data}}$ and sampled labels $y \sim U\{1, C\}$. Hence, to minimize the above divergence, we must minimize the cross entropy which is consistent with classical learning of discriminative models. However, note that in our case the steps in (13) are fully differentiable, so that the data augmentation itself turns into an implicit term in the unified objective function rather than an explicitly constructed set of examples.

Lastly, we wish to re-introduce the entropic regularization for implicit data augmentation. Note, the entropic filter can be interpreted as a hard threshold on $H(p(\hat{y}|x))) < \mathcal{T}$. Here the Langevin dynamics $q_T$ on $z$ maximizes the logit $F_\alpha(z)[y]$, i.e. minimizes $H(p(\hat{y}|x)))$, for which the Wiener process materialized in the noise term $\sqrt{2Ts}\epsilon_k$ with temperature $T$ introduces randomness, or, smoothens the energy potential such that the dynamics converges towards the correct stationary distribution. High temperature $T$ leads to Brownian motion, while low $T$ leads to gradient descent. We realize that $T$ controls $H(p(\hat{y}|x)))$ as it may be interpreted as a soft or stochastic relaxation of $\mathcal{T}$. That is, we can express the entropic filter in terms of the temperature $T$ of $q_T$ and only need to lower $T$ to obtain synthesized samples with associated low entropy in the class logits.

## 3 Experiments: Learning data augmentation

We evaluate our method on standard semi-supervised learning benchmarks for image data. Specifically, we use the street view house numbers (SVHN) [8] data-set with $1,000$ labeled images and $64,932$ unlabeled images. The inference network is a standard Wide ResNet [14]. The generator network is a standard 4-layer de-convolutional network as regularly used in DC-GAN. The energy-based model is a fully connected network with 3 layers. Adam [3] is adopted for optimization with batch-sizes $n = m = l = 100$. The models are trained for $T = 1,200,000$ steps with augmentation after $T_a = 600,000$ steps. The short-run MCMC dynamics in (6) is run for $T_{LD} = 60$ steps.

183 At iteration $T_a$, we take $L$ class conditional samples from the generator with an equal amount of
184 samples ($L/10$ for each digit). We filter conditional samples based on $\mathcal{H}$ as described in Section 2.3
185 for which the threshold $\mathcal{T} = 1\mathrm{e}{-6}$ was determined by grid search. Next, we run CRAIG on the
186 generated samples to keep a subset of 10% of the samples. For these additional examples, we compute
187 the soft-max cross-entropy gradient with per-example weights obtained by CRAIG and update the
188 model with step size $\eta_3 < \eta_2$ or a loss coefficient of $0.1$ to weaken the gradient of $\mathcal{L}_{\mathcal{H}}^{+}$ relative to
189 the original labeled data $\mathcal{L}$. Additionally, for every $10,000$ iteration, we rerun CRAIG to choose an
190 updated subset of generated samples.

| Method | $L$ | | | | |
| | 0 | 10,000 | 40,000 | 100,000 | 200,000 |
| --- | --- | --- | --- | --- | --- |
| Baseline | $92.0 \pm 0.1$ | $88.1 \pm 0.1$ | - | - | - |
| $\mathcal{H}$ | - | $93.5 \pm 0.1$ | $93.8 \pm 0.1$ | - | - |
| $\mathcal{H}$ & CRAIG | - | $93.0 \pm 0.1$ | $93.5 \pm 0.1$ | $93.9 \pm 0.1$ | $93.9 \pm 0.1$ |
| $\mathcal{H}$ & CRAIG & PL | - | - | $94.5 \pm 0.1$ | - | - |

Table 1: Test accuracy with varied number of conditional samples $L$ on SVHN [8].

191 Table 1 depicts results for the test accuracy on SVHN for a varied number of conditional samples $L$.
192 First, we learned the model without data augmentation as a baseline. Then, we draw $L$ conditional
193 samples without an entropic filter and observe worse classification performance. As described earlier,
194 we introduce the entropic filter $\mathcal{H}$ to eliminate conditional samples of low quality which leads to a
195 significant improvement in classification performance with increasing $L$. Finally, we combine both
196 the entropic filter $\mathcal{H}$ and coreset selection by CRAIG to further increase $L$. For $L = 10,000$ there
197 is a significant improvement in classification accuracy when introducing CRAIG, which however
198 decreases with increasing $L$. Lastly, to further boost accuracy we pseudo-label unlabeled data points
199 from the SVHN data-set using the latent classifier. We reject data points whose entropy over the
200 latent classifier is above $10^{-6}$.

## 4  Conclusion

202 In the setting of semi-supervised learning, we have investigated the idea of combining generative
203 models with a coreset selection algorithm, CRAIG. Such a combination is appealing as a generative
204 model can in theory sample an infinite amount of labeled data, while a coreset algorithm can reduce
205 such a large set to a much smaller informative set of synthesized examples. Moreover, learned
206 augmentation is useful as many discrete data modalities such as text, audio, graphs, and molecules do
207 not allow the definition of hand-crafted semantically invariant augmentations (such as rotations for
208 images) easily.

209 We illustrated that a naive implementation of this simple result deteriorates the performance of the
210 classifier in terms of accuracy over a baseline without such data augmentation. The underlying issue
211 here was isolated to being related to the Shannon entropy in the predicted logits over classes for a
212 synthesized example. High entropy indicates samples with low visual fidelity or wrong class identity,
213 which may confuse the discriminative component of the model and lead to a loop in which uncertainty
214 in the predictions leads to worse synthesis. In the first attempt, we constraint the class entropy in the
215 set of augmented examples by taking a subset of the generated data-set with a hard threshold on the
216 Shannon entropy. This resulted in significant empirical improvement of classification accuracy of
217 two percentage points on SVHN. Moreover, we introduced pseudo labels which further improved
218 performance.

219 Then, we show that the latent energy-based model with symbol-vector couplings has conditional
220 distributions for end-to-end training of learned augmentations readily available. We formulate learned
221 data augmentation as the KL-divergence between two known conditional distributions, show the
222 relation to cross-entropy, and relax the entropy regularization into the temperature of the associated
223 Langevin dynamics. This not only allows learning data augmentations as an alteration of the learning
224 objective function but also paves the way toward a theoretical analysis.

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

## A  Algorithms

---

**Algorithm 1:** Semi-supervised learning of generative classifier with coreset selection.

---

**input** : Learning iterations $T$, augmentation iteration $T_a$, learning rates $(\eta_0, \eta_1, \eta_2, \eta_3)$, initial parameters $(\alpha_0, \beta_0, \phi_0)$, observed unlabelled examples $\{x_i\}_{i=1}^{M}$, observed labelled examples $\{(x_i, y_i)\}_{i=M+1}^{M+N}$, unlabelled, labelled and augmented batch sizes $(n, m, l)$, number of augmented samples $L$, entropy threshold $\mathcal{T}$, and number of Langevin dynamics steps $T_{LD}$.

**output :** $(\alpha_T, \beta_T, \phi_T)$.

**for** $t = 0 : T - 1$ **do**

    1. **Mini-batch**:
    Sample $\{x_i\}_{i=1}^{m} \subset \mathcal{U}$, $\{x_i, y_i\}_{i=m+1}^{m+n} \subset \mathcal{L}$, and $\{x_i, y_i\}_{i=m+n+1}^{m+n+l} \subset \mathcal{L}_{\mathcal{H}}^{+}$.
    2. **Prior sampling**:
    For each unlabelled $x_i$, initialize a Markov chain $z_i^{-} \sim q_\phi(z|x_i)$ and update by MCMC with target distribution $p_\alpha(z)$ for $T_{LD}$ steps.
    3. **Posterior sampling**:
    For each $x_i$, sample $z_i^{+} \sim q_\phi(z|x_i)$ using the inference network and reparameterization trick.
    4. **Unsupervised learning of prior model**:
    $\alpha_{t+1} = \alpha_t + \eta_0 \frac{1}{m} \sum_{i=1}^{m} [\nabla_\alpha F_{\alpha_t}(z_i^{+}) - \nabla_\alpha F_{\alpha_t}(z_i^{-})]$.
    5. **Unsupervised learning of inference and generator models**:
    $\psi_{t+1} = \psi_t + \eta_1 \frac{1}{m} \sum_{i=1}^{m} [\nabla_\psi [\log p_{\beta_t}(x|z_i^{+})] - \nabla_\psi \mathrm{KL}(q_{\phi_t}(z|x_i) \| p_0(z)) + \nabla_\psi [F_{\alpha_t}(z_i^{+})]$.
    6. **Supervised learning of prior and inference model**:
    $\theta_{t+1} = \theta_t + \eta_2 \frac{1}{n} \sum_{i=m+1}^{m+n} \sum_{k=1}^{K} y_{i,k} \log(p_{\theta_t}(y_{i,k}|z_i^{+}))$.
    7. **Augment at iteration $T_a$**:
    $\mathcal{Z}_{\mathcal{T}} = \{z_i \sim p(z|y_i) \mid \mathcal{H}(z_i) < \mathcal{T}, i = M + N, \dots, M + N + L\}$,
    $\mathcal{L}_{\mathcal{H}}^{+} = \{(x_i \sim p_\beta(x|z_i), y_i) \mid i = M + N, \dots, M + N + L\}$.
    8. **Approximate the gradient below with CRAIG after iteration $T_a$ according to (12)**:
    $\theta_{t+1} = \theta_{t+1} + \eta_3 \frac{1}{n} \sum_{i=n+m+1}^{m+n+l} \sum_{k=1}^{K} y_{i,k} \log(p_{\theta_t}(y_{i,k}|z_i^{+}))$.

---

## B  Learning the model with variational inference

Given a data point in the unlabeled set, $x \in \mathcal{U}$, the the log-likelihood $\log p_\theta(x)$ is lower bounded by the evidence lower bound (ELBO),

$$\mathrm{ELBO}(\theta) = \mathrm{E}_{q_\phi(z|x)}[\log p_\beta(x|z)] - D_{KL}[q_\phi(z|x) \| p_\alpha(z)] \tag{19}$$

where $\theta = \{\alpha, \beta, \phi\}$ is overloaded for simplicity and $q_\phi(z|x)$ is a variational posterior, an approximation to the intractable true posterior $p_\theta(z|x)$.

For the prior model, the learning gradient for an example is

$$\nabla_\alpha \mathrm{ELBO}(\theta) = \mathrm{E}_{q_\phi(z|x)}[\nabla_\alpha f_\alpha(z)] - \mathrm{E}_{p_\alpha(z)}[\nabla_\alpha f_\alpha(z)] \tag{20}$$

where $f_\alpha(z)$ is the negative free energy defined in equation (4), $\mathrm{E}_{q_\phi(z|x)}$ is approximated by samples from the variational posterior and $\mathrm{E}_{p_\alpha(z)}$ is approximated with short-run MCMC chains [9] initialized from the variational posterior $q_\phi(z|x)$.

Let $\psi = \{\beta, \phi\}$ collects parameters of the inference and generation models, and the learning gradients for the two models are,

$$\nabla_\psi \mathrm{ELBO}(\theta) = \nabla_\psi \mathrm{E}_{q_\phi(z|x)}[\log p_\beta(x|z)] - \nabla_\psi D_{KL}[q_\phi(z|x) \| p_0(z)] + \nabla_\psi \mathrm{E}_{q_\phi(z|x)} f_\alpha(z) \tag{21}$$

where $D_{KL}[q_\phi(z|x) \| p_0(z)]$ is tractable and the expectation in the other two terms is approximated by samples from the varational posterior distribution $q_\phi(z|x)$.

For one example of labeled data, $(x, y) \in \mathcal{L}$, the log-likelihood can be decomposed $\log p_\theta(x, y) = \log p_\theta(x) + \log p_\theta(y|x)$. While we optimize $\log p_\theta(x)$ as the unlabeled data, we maximize $\log p_\theta(y|x)$ by minimizing the cross-entropy as in standard classifier training. Notice that given the Markov chain assumption $y \to z \to x$, we have

$$p_\theta(y|x) = \int p_\theta(y|z) p_\theta(z|x) dz = \mathrm{E}_{p_\theta(z|x)} p_\theta(y|z) \approx \mathrm{E}_{q_\phi(z|x)} \frac{\exp(F_\alpha(x)[y])}{\sum_k \exp(F_\alpha(x)[k])}. \tag{22}$$

In the last step, the true posterior $p_\theta(z|x)$ which requires expensive MCMC is approximated by the amortized inference $q_\phi(z|x)$.

## C  Related Work

**Data augmentation.** Semi-supervised models with purely discriminative learning mostly rely on data augmentation which exploit the class-invariance properties of images. Pseudo-labels [5] train a discriminative classifier on a small set of labelled data and sample labels for a large set of unlabelled data, which in turns is used to further train the classifier supervised. MixMatch [1] applies stochastic transformations to an unlabeled image and each augmented image is fed to a classifier for which the average logit distribution is sharpened by lowering the soft-max temperature. FixMatch [12] strongly distorts an unlabeled image and trains the model such that the cross-entropy between the one-hot pseudo-labels of the original image and the logits of the distorted image is minimized. Mean teacher [13] employs a teacher model which parameters are the running mean of a student model and trains the student such that a discrepancy between teacher and student predictions of augmented unlabeled examples is minimized. Virtual Adversarial Training (VAT) [7] finds an adversarial augmentation to an unlabeled example within an $\epsilon$-ball with respect to some norm such that the distance between the class distribution conditional on the unlabeled example and the one on the adversarial example is maximized.

The methods of MixMatch, FixMatch and Mean teacher rely on pre-defined data augmentations, which are readily available in the modality of images as the semantic meaning is invariant to transforms such as rotation or flipping, but are difficult to construct in modalities such as language or audio modalities. Our method is agnostic to the data modality. Pseudo-labeling is closely related in that labels are sampled given unlabeled examples, whereas our method samples examples given labels. VAT is close to our method as it is modality agnostic and leverages the learned model to sample labeled examples, albeit of "adversarial" nature while our samples are "complementary." DAPPER is closest to our method as it employs a generative model to augment the data-set, but it misses the coreset reduction.

