# OpenReview forum: "Generating High Fidelity Synthetic Data via Coreset selection and Entropic Regularization"
_NeurIPS.cc/2022/Workshop/SyntheticData4ML — Neurips 2022 SyntheticData4ML_

### Official Review · Reviewer_PspK · 2022-10-19
**An interesting data augmentation method based on coreset and entropic filter but lack of evidence**

**Rating:** 4
**Confidence:** 3

**Review:**

- Quality: well organized paper, interesting idea for data augmentation.
- Clarity: some concepts and equations need more explanations.
- Originality: the idea of combining coreset identification and entropic filter is new, but the technical contribution of this paper is not strong.
- Significance: the considered problem of high quality data augmentation is very important in the area of synthetic data.

The data augmentation problem in semi-supervised setting is considered in this paper, and the authors proposed a new approach via the combination of coreset approach and an entropic filter to select synthetic samples of high fidelity for data augmentation. This paper is properly structured and in general well written. However, the method description is not quite clear, and explanations for certain notations are missing. For instance, the choice of dynamical model in eq.6 is not discussed, and no explanation for variable s is given.

The proposed method utilizes an existing coreset discovery algorithm CRAIG together with a novel data filter based on sample entropy. The technical contribution in this paper is not strong, and the result in Fig. 1 does not demonstrate obvious advantage of the proposed method. The benchmark results in Table 1 are not very convincing as no existing data augmentation approaches are evaluated as baselines.

The authors are suggested to improve the general clarity of their method section and include some existing data augmentation methods as baselines.

---

### Official Review · Reviewer_Gmya · 2022-10-20

**Rating:** 5
**Confidence:** 3

**Review:**

This work combines the VAE and energy-based model.
During data augmentation, they first use an energy-based model (or prior model) to generate the latent variables based on labels and then use VAE's decoder to generate the fake data given latent variables. If it's the semi-supervised learning setting, they can feed the VAE encoder with unlabelled real data to get additional latent variables.
On top of this pipeline, they apply two tricks to filter high-fidelity images 1) they use Coreset to select the most representative images from the generated images, and 2) they filter out the latent variables with high entropy. High entropy means that the classifier is less certain about the label of the image, and thus the image is more likely to be fake. The entropy selection idea is novel to me, but other parts are from existing works.

Clarity: There is a range of typos or errors in the paper.

- In the crutial equation (8), the entropy should be $-\sum_y p(y|z) \log p(y|z)$, otherwise it is not entropy.

- In line 102, the $p(x|z) p(z|y)$ should add an intergral over $z$.

- No definition of $\Sigma_L'$ in eq (12)

- In eq (15), there shouldn't be a $p(z|y)$ in the expectation.

- The result in Figure 1b looks suspicious. It only includes digits 9.

Overall, the model includes three networks, and the learning process is relatively complex.
From the result of the paper, the improvement is not obvious compared to the baseline without data augmentation. The highest improvement is 2%. There is even a drop in accuracy due to the use of CRAIG. From Figure 1, I cannot see too many differences between plots (a) and (b). Plot b also involves a couple of low-fidelity images, especially in the last row.

---

### Official Review · Reviewer_3PTC · 2022-10-20
**Easy to follow but novelty is limited and experiments are not convincing**

**Rating:** 5
**Confidence:** 3

**Review:**

This paper suggests to use a latent energy based model [1] to select high quality samples when generating synthetic data and also use coreset selection [2] to further filter out some of the samples. The paper is clear and easy to follow. However, the novelty seems to be limited. Section 2.1 and 2.2 seem to be a summarization of the method proposed in [1]. Here, the main contribution of this paper is that they propose to filter out the samples with high entropy via an arbitrary threshold chosen by grid-search. It is not clear how sensitive the model is to this parameter, as only results for threshold = 1e-6 is reported. Additionally, one potential issue might be that the entropic filter might prefer to select a few classes that it is the most confident about. For example, in Figure 1b, most images selected are all number "9"s. The application of coresets also seems to be directly inherited from prior works.

Moreover, the authors claim that coresets can be used together with the proposed entropic filter. In the caption for Table 1, the authors state that "For L = 10, 000 there is a significant improvement in classification accuracy when introducing CRAIG, which however decreases with increasing L. " However, in Table 1, applying both techniques leads to worse results (93.0 vs 93.5 for L = 10,000; 93.5 vs. 93.8 for L = 40,000). The proposed framework is only tested on one dataset, and many entries in the table is missing.

[1] https://proceedings.neurips.cc/paper/2020/file/fa3060edb66e6ff4507886f9912e1ab9-Paper.pdf
[2] https://arxiv.org/pdf/1906.01827.pdf

---

### Official Review · Reviewer_TiPX · 2022-10-20
**Neat idea, well written paper**

**Rating:** 9
**Confidence:** 4

**Review:**

The paper discusses how to learn high quality subset of the synthetic data that will be most useful in the context of data augmentation. Well-written paper - good analysis of related work,  very high level of detail, solid experiments. recommend to accept.

---

### Meta-Review · Area_Chair_yDrw · 2022-10-20

**Recommendation:** Accept

**Review:**

The authors introduce an interesting idea that generates high-quality augmented data. The idea is novel and should be of great interest to the community, but the exposition could be improved to better communicate the technical contributions of the paper. I would highly recommend the authors take in the comments provided by the reviewers (esp. around baselines + experiments) to turn this into a stronger paper.